# Novel Pathogenic Mutations Identified from Whole-Genome Sequencing in Unsolved Cases of Patients Affected with Inherited Retinal Diseases

**DOI:** 10.3390/genes14020447

**Published:** 2023-02-09

**Authors:** Hafiz Muhammad Jafar Hussain, Meng Wang, Austin Huang, Ryan Schmidt, Xinye Qian, Paul Yang, Molly Marra, Yumei Li, Mark E. Pennesi, Rui Chen

**Affiliations:** 1Department of Molecular and Human Genetics, Baylor College of Medicine, Houston, TX 77030, USA; 2Human Genome Sequencing Center, Baylor College of Medicine, Houston, TX 77030, USA; 3Department of Ophthalmology, Casey Eye Institute, Oregon Health & Science University, Portland, OR 97239, USA

**Keywords:** inherited retinal diseases, whole-genome sequencing (WGS), targeted gene panels, whole-exome sequencing, deep intronic mutations

## Abstract

Inherited retinal diseases (IRDs) are a diverse set of visual disorders that collectively represent a major cause of early-onset blindness. With the reduction in sequencing costs in recent years, whole-genome sequencing (WGS) is being used more frequently, particularly when targeted gene panels and whole-exome sequencing (WES) fail to detect pathogenic mutations in patients. In this study, we performed mutation screens using WGS for a cohort of 311 IRD patients whose mutations were undetermined. A total of nine putative pathogenic mutations in six IRD patients were identified, including six novel mutations. Among them, four were deep intronic mutations that affected mRNA splicing, while the other five affected protein-coding sequences. Our results suggested that the rate of resolution of unsolved cases via targeted gene panels and WES can be further enhanced with WGS; however, the overall improvement may be limited.

## 1. Introduction

Inherited retinal diseases (IRDs) are among the most severe and irreversible causes of blindness for millions of patients. It has been shown through the use of high-throughput sequencing technologies that IRDs exhibit a wide range of phenotypic and genetic heterogeneity with more than 280 genes and loci associated with autosomal-recessive, autosomal-dominant, X-linked, and mitochondrial inheritance (https://web.sph.uth.edu/RetNet/sum-dis.htm accessed on 10 January 2023). Due to the high heterogeneity in both genetic and clinical phenotypes, it is crucial to identify the pathogenic mutations for each IRD patient to provide improved diagnosis, prognosis, and genetic counseling. Consequently, the utilization of high-throughput sequencing techniques has become the standard of care for IRDs in recent years [1,2,3,4]. On the other hand, the retina is a favorable target for developing gene therapy due to its small volume, direct visibility, and immuno-privileged environment, as well as the various sensitive procedures that can be performed to assess its function [5]. As a result, with the rapid development of gene therapy that targets IRD diseases, accurate diagnoses at the molecular level are essential in matching patients with proper personalized treatment [5].

Genomic analysis is used as the diagnostic standard to distinguish among genes with diverse phenotypes that are attributed to variants. Previously, targeted gene panels and whole-exome sequencing (WES) were used for the detection of mutations in coding sequences and nearby splicing sites [6,7,8]. Whole-genome sequencing (WGS) affords significant advantages over traditional gene panels and WES because it provides more even and complete coverage for the entire genome by covering every base in both the coding and noncoding regions [9,10]. A recent study by Ellingford et al. indicated that the causal variant detection rate by WGS was significantly higher than that of targeted panels in IRD patients [7]. A study by Keren et al. demonstrated a 56% variant detection rate in IRD patients, which was considerably higher than the rate detected by WES [11]. Aziz et al. described that the proportion of false-positive variants for WES was greater (78%) than for WGS (17%) [12]. These studies not only emphasized the utility of WGS in the diagnosis of unresolved IRD cases following techniques such as targeted gene panels and WES, but several groups also identified novel mutations to establish underlying genetic causes for cryptic IRDs [8,11,13,14]. However, the increase in the diagnostic rate by WGS varied significantly among previous studies and ranged from a few percent to 24%. In this study, we examined the contribution of WGS in the diagnosis of a cohort of 311 IRD patients who were negative in targeted panel sequencing and WES.

## 2. Materials and Methods

### 2.1. Subjects

All probands in our cohort were clinically diagnosed with inherited retinal diseases by a qualified panel of ophthalmologists. Genetic counseling and DNA analysis via blood collection were conducted following the provision of written informed consent by the participants. DNA samples from probands were extracted using Qiagen blood genomic DNA extraction kits (Qiagen, Hilden, Germany). This study adhered to the tenets of the Declaration of Helsinki and was approved by the institutional review boards at every affiliated institution.

### 2.2. Whole-Genome Sequencing of IRD Patients

In order to identify pathogenic mutations, gene panel testing was performed for all patients in this cohort. Further in-depth analysis was conducted on patients who were not able to be assigned a certain molecular diagnosis. WGS results were processed at the Human Genome Sequencing Center at Baylor College of Medicine using a modified pipeline from our previous WES and WGS analyses [13,15]. WGS sequencing reads were aligned briefly with the Burrows–Wheeler Alignment (BWA) human genome assembly (hg19) [16], whereas single-nucleotide variants (SNVs) and insertion–deletion variants (INDELs) were identified using GATK4. To eliminate frequently occurring variants that were not likely to cause IRDs, a 0.5% population-frequency threshold was set. Coding region SNVs and INDELs were annotated with ANNOVAR and compared to the dbNSFP 3.5a database, while the remaining variants’ conservation was estimated in accordance with the UCSC Genome Browser’s phastCons.hg19.100way [17]. Prediction of the effect of coding variants was performed using CADD v1.3 [18,19].

WGS-consolidated SNVs from all patients were annotated and filtered according to the genomic alteration with a custom pipeline, and the prediction of intronic variant effects was performed with SpliceAI (spliceai-1.2.1) [20]. Each variant was given a score between 0 and 1 by the in silico variant-prediction engine SpliceAI; a higher score indicated greater confidence that the variant affected the splicing. Potential splicing variants were chosen with a prediction cutoff score of 0.5 after being restricted to previously reported IRD genes and analyzed in accordance with known inheritance patterns. One hit of the splicing variant was required for IRD genes associated with dominant or X-linked hemizygous diseases, while an additional coding or splice-affecting variant was needed for genes that were linked to recessive diseases (Appendix A). In order to assess only deep intronic splicing variants by removing those that were too close to canonical splice sites, these candidate variants were also filtered according to the distance from exon–intron junctions at >10 bp.

### 2.3. In Vitro Validation of Novel Intronic Variants

Following the prioritization of the variants, a minigene reporter assay (RHCglo minigene) was performed to analyze the splicing effects [21]. Site-directed mutagenesis was conducted with the vector using the WT amplicon for patients with greater than one mutation in the PCR-amplified region. These products were then cloned into the RHCglo vector, and the effects of splicing variants were assessed following the transfection of plasmids into HEK-293 cells and a subsequent RT-PCR assay [22]. Quantification of the DNA band intensity was performed using the ImageJ Gel Analysis program (https://imagej.nih.gov/ij/, accessed on 14 January 2022); the primer sequences for RT-PCR are listed in Appendix A.

## 3. Results

To explore the genetic variants in IRD patients, we analyzed the WGS data from a cohort of 311 IRD patients whose cases remained unsolved via panel sequencing or WES. Upon filtering as described in the Section 2, a total of nine candidate pathogenic variants (Table 1) in six IRD patients were identified (Table 2). Among these nine variants, three had been previously reported while the remaining six were novel. Moreover, five variants were exonic (one missense, three frameshift, and one indel), and four variants were deep intronic and found beyond 50 bp of the exon–intron boundary.

### 3.1. Validation of Novel Intronic Cryptic Splicing Variants

Among the identified variants in the IRD patients, two were novel intronic splicing variants that were predicted by SpliceAI to create both cryptic donor and acceptor sites (Table 1). To validate the SpliceAI prediction for the novel intronic splicing variants, we performed a minigene assay and revealed the functional impact of the identified candidate variants on mRNA splicing. An RHCglo minigene system was used to perform an in vitro functional splicing assay [20]. The results of the in vitro assay were consistent with the in silico prediction. Moreover, for the novel candidate splicing variants, the RT-PCR produced new bands as predicted by SpliceAI with different lengths as compared to the wild type. The details of each patient and the mutant alleles are described below.

A 34-year-old Caucasian female (MEP-123) was diagnosed with retinitis pigmentosa (RP) (Figure 1A) without a family history of the disease (Figure 1A and Table 2). The father (I:1) of the proband was affected by hearing loss. WGS identified two heterozygous variants in *EYS*, mutations in which led to autosomal recessive RP. One variant was a splicing variant (NM_001142800.2: c.2259+3291G>T) in intron 14 of *EYS*, and the second variant (NM_001142800.2:216delA) was a deletion of one nucleotide in exon 4 (Figure 1B and Table 1). Both variants were novel and are rare in the population (0.00003188 and absent) as estimated according to public databases such as gnomAD. The splicing variant c.2259+3291G>T was predicted to create a novel splicing donor site downstream of exon 14 causing an out-of-frame-insertion of a new cryptic exon of 59 bp between exons 14 and 15 (Figure 1C). Similarly, the frameshift introduced by the second variant led to premature stop that was likely to result in nonsense-mediated mRNA decay and therefore a null allele. EYS is located in the connecting cilium of the photoreceptor, and earlier studies proved that the disruption of EYS in zebrafish can cause an abnormality in the localization of the outer-segment proteins and degeneration of the photoreceptors, thus supporting its important role in the retinal architecture [26,27,28]. Taken together, these variants were likely pathogenic and caused RP in the patient.

Based on the in vitro minigene assay, the c.2259+3291G>T variant produced a major and intense band (Figure 1D). This major band was caused by the identified variant and was composed of the aberrant transcript of exon 14 and the addition of 58 bp downstream, which exactly matched the in silico prediction, while a minor and light band was identified as the wild type (Figure 1D). The mutant isoform carried the original exon and a cryptic exon of 58 bp, which perfectly matched the in silico prediction as confirmed by Sanger sequencing (Figure 1E). Thus, the c.2259+3291G>T variant is likely a pathogenic variant that causes retinitis pigmentosa.

A 50-year-old Caucasian male patient (MEP-395) was diagnosed with choroideremia (Figure 2A). He had no family history of the disease. His father (II:2) and paternal uncle (II:1) had partial hearing loss, and his mother (II:3) had slight night blindness (Figure 2A). WGS identified a hemizygous splicing variant (c.117-962G>C) in intron 2 of *CHM* as shown in the IGV plot (Figure 2B). The splicing variant c.117-962G>C was predicted to create a novel splicing donor site downstream of exon 2 causing an out-of-frame-insertion of a new cryptic exon of 115 bp between exons 2 and 3 *(*Figure 2C). The cryptic exon contained an early stop codon downstream of the cryptic acceptor splice site that resulted in the generation of a premature stop codon downstream of amino acid position 62 out of a total of 653 amino acids in the wild-type protein. *CHM* encodes Rab escort protein 1 (REP1), which is crucial for vesicle trafficking. In humans, any abnormality in *CHM* can cause the characteristic clinical phenotype choroideremia, which is a progressive centripetal retinal degenerative disease that appears only to affect the retinal pigment epithelium (RPE) layer [29,30,31]. All the functionally important domains of REP1 lie downstream of the mutant site, which may result in a very short REP1 protein with loss of all the functional domains. Consistent with the predicted results, the in vitro minigene assay for the c.117-962G>*C* variant produced two bands (Figure 2D). The relative band intensity of the mutant and wild type was 40% and 60%, respectively (Figure 2D). Of the two bands observed, the prominent band was caused by the identified variant and was composed of the atypical transcript of exon 2 and an addition of 114 bp downstream, which matched the in silico prediction, while the minor band was determined to be the wild type (Figure 2D). The minor isoform contained the original exon, while the major isoform consisted of the original exon plus a cryptic exon of 114 bp, which was confirmed by Sanger sequencing and exactly matched the in silico prediction (Figure 2E). Taken together, the c.2259+3291G>T variant was likely a pathogenic variant that caused choroideremia in patient MEP-395.

### 3.2. Patients Carrying Novel Coding Pathogenic Mutations

A 19-year-old Asian male (MEP-398) was diagnosed with RP with a paternal family history of minor vision problems (Figure 3A). Father (III:2) of the proband had partial hearing loss. WGS identified two heterozygous variants (NM_014714.4: c.1487C>T and c.1250_1271dup) in exon 13 and exon 11 of *IFT140,* respectively (Appendix A). The nonsynonymous sequence change caused by the coding variant c.1487C>T replaced a highly conserved threonine residue with methionine at codon 496 of the IFT140 protein (p.T496M), which was deemed to be novel and rare (0.000606) in the population database gnomAD. Moreover, multiple in silico algorithms predicted this variant to have a deleterious effect (GERP++ rank score = 0.89; REVEL score = 0.95). Another heterozygous variant was a duplication that was predicted to cause an early stop codon (p.S425Gfs*66) in *IFT140*. This variant was also novel and is extremely rare in the population: it was absent in the genome-sequencing database gnomAD. If translated, this variant is predicted to either produce a very short protein of only 490 instead of 1462 amino acids or cause nonsense-mediated decay (NMD) of mRNA. IFT140 is a subunit of IFT-A that plays a crucial role in the maintenance and development of the outer segments [32]. Mutations in *IFT140* lead to autosomal recessive retinitis pigmentosa (ASRP) [33]. Taken together, both variants were likely to be pathogenic in the patient.

MEP-662 and MEP-663 were two affected Asian siblings that were diagnosed with cone–rod dystrophy (Figure 3A,C,D). WGS of the patient’s DNA identified compound heterozygous variants (NM_004928.3:c.634_635del and c.351_352insACCCTGCCGCGC) in both siblings in exon 6 and exon 4 of *CFAP410*/*C21orf2*, respectively (Appendix A). One of the variants (c.634_635del) was a novel frameshift variant that was predicted to cause an early stop codon at amino acid p.R212GfsTer, while the second variant (c.351_352insACCCTGCCGCGC) was a recurrent pathogenic mutation [23]. *CFAP410*/*C21orf2* is a ubiquitous protein that has been associated with different cellular functions such as DNA damage repair, the regulation of cell morphology, and cytoskeletal organization [34,35]. Previous studies reported that variants in *CFAP410*/*C21orf2* can cause retinitis pigmentosa and cone–rod dystrophy [36]. Moreover, based on the predictive effects of both variants, these were likely the disease-causing mutations in the patients.

### 3.3. Patient with Reported Pathogenic Mutations

MEP-082 was a 46-year-old Caucasian male who was diagnosed with ABCA4-related retinopathy (Figure 3A,E) and had no family history of retinal disease. However, his father (II:2) and a paternal uncle (II:1) were affected by amyotrophic lateral sclerosis (ALS) (Figure 3A). The genetic analysis via WGS identified two heterozygous known variants (NM_000350.3; c.5196+1137G>A and c.1555-2745A>G) in intron 36 and 11 of *ABCA4,* respectively. Both variants are rare in the population and had a very low frequency (0.000264 and 0.000397, respectively) in the genome sequencing database gnomAD (Appendix A). Both variants were previously reported as pathogenic mutations. The identified variant (c.5196+1137G>A) was present in *trans* with c.5196+1216C>A by haplotyping in a patient affected with Stargardt disease [24]. Interestingly, the second variant (c.1555:2745A>G) was detected along with the first variant (c.5196+1137G>A) and another variant (p.C54Y) in the same gene (*ABCA4*) in a patient affected with Stargardt disease [25]. In short, our identified variants (NM_000350.3; c.5196+1137G>A and c.1555-2745A>G) were reported previously and caused the disease in our patient.

## 4. Discussion

In this study, WGS was performed in 311 probands whose mutations remained unknown after targeted panel sequencing or WES. Following systematic screening criteria of causal variants in known genes, nine causative variants were identified. Out of nine variants, six were novel and three were previously described. Among the novel variants, one was missense, three were frameshift, and two were deep intronic splicing variants.

The two novel intronic splicing variants (the c.2259+3291G>T variant in *EYS* and the c.117-962G>C variant in *CHM*) activated cryptic donor and acceptor splice sites close to the mutation sites and thereby resulted in cryptic exon inclusion. Both novel splicing variants were deleterious for the following reasons: (1) they were rare in the population; (2) the predictions were further validated by a minigene assay; and (3) the clinical phenotypes were consistent with the genotypes. It is worth noting that two out of four of the intronic variants that we identified were novel and had not been reported previously, which suggested that a considerable portion (50%) of cryptic splicing mutations remain undiscovered by conventional sequencing techniques [13]. Most of the exonic variants (75%, 3/4) were frameshift, which was missed by targeted panel sequencing or WES due to various reasons such as out-of-date gene panels, the inadequacy of the annotation pipeline, or a lack of knowledge and evidence regarding the interpretation of IRD-associated genes at the time of the targeted panel sequencing or WES data analysis [37].

In previous studies, an improved mutation-detection rate via WGS was described compared to targeted panels and WES, although the improvement varied between a few percent and 24% [1,2,3,4,7,11,37,38,39,40]. It was found that the WGS was superior in the detection of SVs, variants in regulatory regions, and variants in GC-rich regions compared to WES [11]. In another recent study, Fadaie Z. et al. detected disease-causing variants in 24 out of 100 unsolved IRD cases, which was the highest percentage of variant detection by WGS to date [37]. However, it is worth noting that a significant portion of mutations missed by capture sequencing mapped to coding regions or near canonical splicing sites, which indicated that the improvement in WGS was not simply due to the improvement in the sequencing coverage. Factors that can influence the differences between the detection rates include the degree of prescreening performed, the panel design utilized, and the data analysis. In the cohort investigated in this study, a customized gene panel that included known deep intronic mutations was used in the initial screening. In addition, patients with SVs and large CNVs were not included in this report because they were described in our previous study [40]. As result, novel deep intronic mutations and small indels missed by our previous screen via panel sequencing and WES were only detected in 1.93% (6/311) of the unsolved patient cohort.

This study illustrated that while improvement is limited, WGS can improve the diagnostic rate for unsolved IRD cases. It is plausible that long-read WGS could provide a higher diagnostic yield by detecting causal variants that are missed by the short WGS technologies [37]. To further increase the diagnostic rate, the contribution of other types of mutations such as those that affect gene expression regulatory elements need to be systematically assessed. WGS coupled with a functional assay of candidate noncoding variants is essential to identify and confirm the pathogenicity of these noncoding mutations.

In conclusion, WGS offered some advantages over gene panel sequencing and WES in detecting deep intronic mutations, SVs and CNVs, and indels, therefore increasing the yield, although the improvement was limited. As a result, the majority of IRD cases remain unsolved despite the introduction of WGS. Several possibilities to increase the detection rate warrant further investigation; these include cryptic splicing sites missed by current prediction tools, SVs, gene regulatory elements and duplication regions that are hard to detect with short-read sequencing technology, and novel disease-associated genes.

## Figures and Tables

**Figure 1 genes-14-00447-f001:**
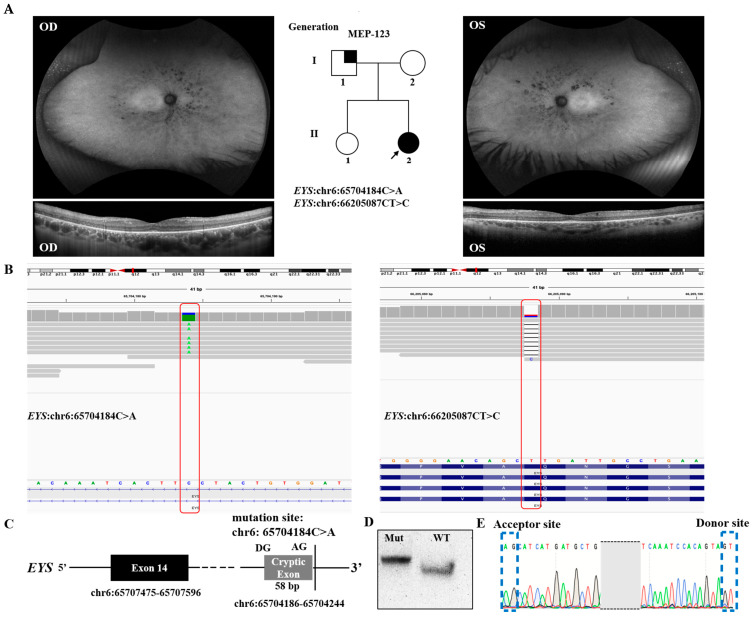
Clinical and functional validation of intronic splicing variant of *EYS* in MEP-123, who was affected with retinitis pigmentosa (RP). (**A**) Pedigree of proband, fundus autofluorescence (AF), and optical coherence tomography (OCT) for MEP-123. Symbols represent: females (circles), males (squares), affected individuals (filled symbols), unaffected individuals (open symbols), proband (arrowhead), and hearing loss (partially filled symbols). (**B**) IGV plots showing the mutant region. A proband (MEP-123) affected with RP had a pathogenic heterozygous deep intronic variant in *EYS* (chr6:65704184C>A). The individual also had a pathogenic frameshift variant NM_001142800.2: c.216delA (p.Q72fs). (**C**) Predictive results of novel splicing variant identified in proband. (**D**) Gel electrophoresis of reverse-transcription PCR (RT-PCR) of novel splicing variant and WT. (**E**) Confirmation of novel donor and acceptor sites of cryptic exon via Sanger sequencing. Blue dotted lines are showing donor and acceptor sites.

**Figure 2 genes-14-00447-f002:**
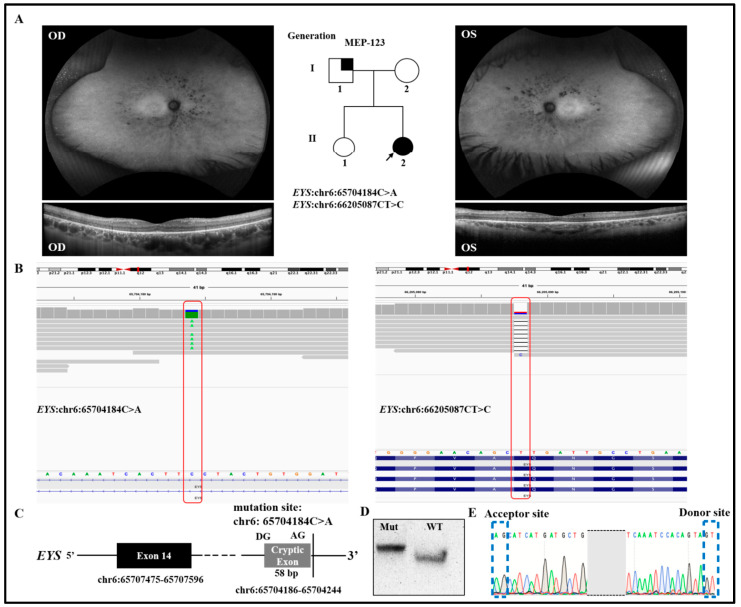
Clinical and functional validation of intronic splicing variant of *CHM* in MEP-395, who was affected with choroideremia. (**A**) Pedigree of proband, fundus autofluorescence (AF), and optical coherence tomography (OCT) for MEP-395. Symbols represent: females (circles), males (squares), affected individuals (filled symbols), unaffected individuals (open symbols), proband (arrowhead), and slight night blindness (star). (**B**) IGV plot showing the mutant region. A male proband (MEP-395) affected with choroideremia had a pathogenic hemizygous deep intronic variant in *CHM* (chrX:85237775C>G). (**C**) Predictive results of novel splicing variant identified in proband. (**D**) Gel electrophoresis of reverse-transcription PCR (RT-PCR) of novel splicing variant and WT. (**E**) Confirmation of novel donor and acceptor sites of cryptic exon via Sanger sequencing. Blue dotted lines are showing donor and acceptor sites.

**Figure 3 genes-14-00447-f003:**
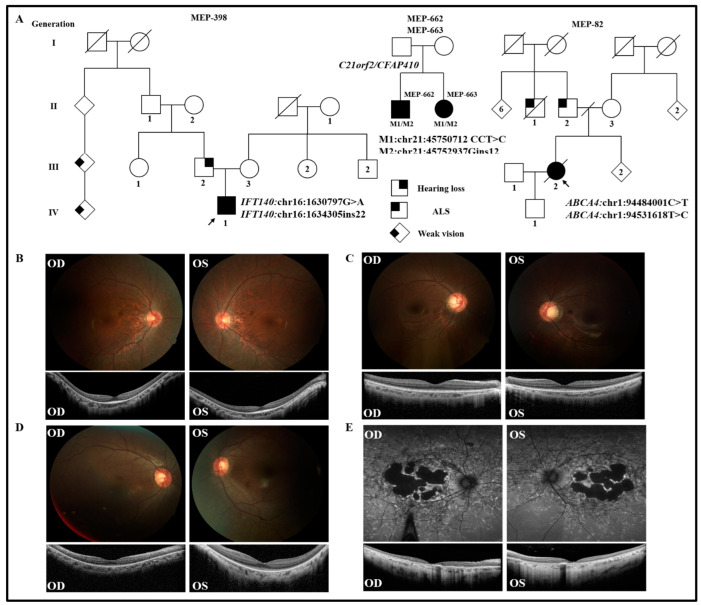
Clinical data of probands affected with IRD. (**A**) Pedigrees of probands (MEP-398 and MEP-662/663) and MEP-082. Symbols represent: females (circles), males (squares), unknown sex (diamonds), affected individuals (filled symbols), unaffected individuals (open symbols), numbers of siblings (numbers in symbols), and deceased (oblique line through symbol) (**B**) Fundus autofluorescence (AF) and optical coherence tomography (OCT) of MEP-398. (**C**) AF and OCT of MEP-662. (**D**) AF and OCT of MEP-663. (**E**) AF and OCT of MEP-082.

**Table 1 genes-14-00447-t001:** Identified variants in IRD patients.

Patient ID	Gene	Genomic Variant	cDNA Variant	Protein Variant	Zygosity	Variant Type	Reference
MEP-123	*EYS*	chr6:65704184C>A	NM_001142800.2:c.2259+3291G>T	-	Heterozygous	Splicing	Novel
	*EYS*	chr6: 66205087CT>C	NM_001142800.2:c.216delA	NP_00113627.1:p. A73Lfs*12	Heterozygous	Frameshift	Novel
MEP-395	*CHM*	chrX:85237775C>G	NM_000390.4:c.117-962G>C	-	Hemizygous	Splicing	Novel
MEP-398	*IFT140*	chr16:1630797G>A	NM_014714.4:c.1487C>T	NP_055529.2:T496M	Heterozygous	Missense	Novel
	*IFT140*	chr16:1634305ins22	NM_014714.4:c.1250_1271dup	NP_055529.2:p. S425Gfs*66	Heterozygous	Frameshift	Novel
MEP-662	*C21orf2/* *CFAP410*	chr21:45750712 CCT>C	NM_004928.3:c.634_635del	NP_004919.1:p. R212Gfs	Heterozygous	Frameshift	Novel
	*C21orf2/* *CFAP410*	chr21:45752937Gins12	NM_004928.3:c.351_353dup12	NP_004919.1:p. L118delinsTLPRL	Heterozygous	Insertion	[23]
MEP-663	*C21orf2/* *CFAP410*	chr21:45750712 CCT>C	NM_004928.3:c.634_635del	NP_004919.1:p. R212Gfs	Heterozygous	Frameshift	Novel
	*C21orf2/* *CFAP410*	chr21:45752937Gins12	NM_004928.3:c.352_352dup12	NP_004919.1:p. L118delinsTLPRL	Heterozygous	Insertion	[23]
MEP-082	*ABCA4*	chr1:94484001C>T	NM_000350.3:c.5196+1137G>A	-	Heterozygous	Splicing	[24]
	*ABCA4*	chr1:94531618T>C	NM_000350.3:c.1555-2745A>G	-	Heterozygous	Splicing	[25]

**Table 2 genes-14-00447-t002:** Clinical features of selected IRD patients.

Patient ID	Clinical Diagnosis	Gender	Race	Age (Years)	Age of Onset (Years)	BCVA		Progression	Other Symptoms	Family History
						Right	Left			
MEP-123	RP	F	Caucasian	34	20	20/50-	20/50-	Mild progression of cystoid macular edema, peripheral vision, and night blindness	NA	Isolated case
MEP-395	Choroideremia	M	Caucasian	50	38	20/20-2	20/80	Mild progression of visual field	Night blindness, legally blind	Mother with possible “slight night blindness”
MEP-398	RP	M	Asian	19	9	20/25 + 2	20/20-2	Patient had minimal progression	Nyctalopia, midperipheral scotomas, midperipheral atrophy, reduced ERGs, visual fields, loss of outer retinal structures on OCT, high myopia	Some relatives with weak vision but not a similar phenotype
MEP-662	CRD	M	Asian	3	12	20/70 + 1	20/70+1	Gradual tilting of RNFL over time	Primary congenital glaucoma, subnormal visual acuity, severe cone and mild rod dysfunction	Affected sister with negative family history of similar eye problems
MEP-663	CRD	F	Asian	1	9	20/150	20/150 + 1	Low vision from cone–rod dystrophy	Hyperopia, nystagmus, astigmatism, intermittent exotropia OU, progressive cone–rod dystrophy	Affected brother with negative family history of similar eye problems
MEP-082	ABCA4 related retinopathy	F	Caucasian	46	NA	CF	20/100	Progression of retinopathy OU with foveal sparing	Deceased in January 2022	Family history of ALS but not vision problems

RP, retinitis pigmentosa; CRD, cone–rod dystrophy; ALS, amyotrophic lateral sclerosis; CF, count fingers visual acuity; NA, not available.

## Data Availability

Due to the concerns of patients, all datasets for this manuscript are not publicly available. Requests to access the datasets should be directed to the corresponding author.

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
