# Peer review of "Novel Pathogenic Mutations Identified from Whole-Genome Sequencing in Unsolved Cases of Patients Affected with Inherited Retinal Diseases"

_genes, 2023, doi:10.3390/genes14020447_

Round 1
Reviewer 1 Report
Genetically inherited retinal diseases (IRDs) are the leading cause of blindness in the world. Understanding which causative mutations lead to these diseases is key to gene therapy treatments. Several sequencing techniques are routinely used in clinics to identify these disease genes in patients. Sequencing of human samples against known IRD genes is limited only the known IRD causing genes. Sequencing of the just exons allows for more genomic coverage but still some IRDs remain undiagnosed. This is because several noncoding regions, the intronic region, also plays a crucial role in affecting the mRNA processing and hence the protein translation.
In this work, Hussain et al., are describing the importance of whole genome sequencing (WGS) and the merits it carries in defining cryptic mutations that remain undiagnosed in certain IRDs when using conventional sequencing of just exons (whole exome sequencing (WES)) and the specific gene panel.
The authors performed WGS sequencing and validated some of the mutations using in vitro assays and have found 6 novel mutations that lie in the deep intronic region. The article has good scientific bases and has reported novel mutations that are of importance to diagnose IRD patients. The article however needs several modifications.
The following are my concerns/suggestions:
1] The Figure 1 describes patient MEP 395 and not MEP 123. Moreover, in the figure legend for Fig1, the patient ID is written as MEP-213. If Figure 2 is to be considered as the Fig1, the description of panel D in the results section, says that Exon 14 mutation gives rise to two bands in the gel. The gel image is described to have 2 bands in the mutant lane, but there is only one band. Because of the mix-up in the labeling, the figures were hard to follow.
2] In the description on the second patient on page6, the introduction of GCTase and REP-1 in the body of the text seems abrupt. Can you provide some reference to the normal function of CHM, GCTase, and REP-1 and how they are connected? This will help understand your results better.
3] A sentence or a reference paper describing the normal function of genes (EYS, CHM) that are affected in patients might help understand the importance of that gene better.
Author Response
Dear Miss Mara Pop,
Thank you for the review and the opportunity to revise the manuscript submission ID: 2193605 entitled “Novel pathogenic mutations identified from whole-genome sequencing of unsolved patients affected with inherited retinal diseases
”. We greatly appreciate the effort and time expended by the reviewers to provide us with valuable input to improve our manuscript. We have addressed every comment provided by the reviewers. Below, please find the comments and our responses to each comment.
All changes made to the main text were marked by tracked changes in the “Revision_1-30-2023.docx” files.
Reviewer 1’s comments
Comments to the Author
Genetically inherited retinal diseases (IRDs) are the leading cause of blindness in the world. Understanding which causative mutations lead to these diseases is key to gene therapy treatments. Several sequencing techniques are routinely used in clinics to identify these disease genes in patients. Sequencing of human samples against known IRD genes is limited only the known IRD causing genes. Sequencing of the just exons allows for more genomic coverage but still some IRDs remain undiagnosed. This is because several noncoding regions, the intronic region, also plays a crucial role in affecting the mRNA processing and hence the protein translation.
In this work, Hussain et al., are describing the importance of whole genome sequencing (WGS) and the merits it carries in defining cryptic mutations that remain undiagnosed in certain IRDs when using conventional sequencing of just exons (whole exome sequencing (WES)) and the specific gene panel.
The authors performed WGS sequencing and validated some of the mutations using in vitro assays and have found 6 novel mutations that lie in the deep intronic region. The article has good scientific bases and has reported novel mutations that are of importance to diagnose IRD patients. The article however needs several modifications.
The following are my concerns/suggestions:
Comment 1
The Figure 1 describes patient MEP 395 and not MEP 123. Moreover, in the figure legend for Fig1, the patient ID is written as MEP-213. If Figure 2 is to be considered as the Fig1, the description of panel D in the results section, says that Exon 14 mutation gives rise to two bands in the gel. The gel image is described to have 2 bands in the mutant lane, but there is only one band. Because of the mix-up in the labeling, the figures were hard to follow.
Response
Thank you for pointing this out. We corrected the patient ID in the figure legend. We also corrected the description of Figure 1D in the results part of MEP-123.
Comment 2
In the description on the second patient on page6, the introduction of GCTase and REP-1 in the body of the text seems abrupt. Can you provide some reference to the normal function of CHM, GCTase, and REP-1 and how they are connected? This will help understand your results better.
Response
Thank you for your suggestion. We have described the role and function of every gene and added references for a better understanding.
Comment 3
A sentence or a reference paper describing the normal function of genes (EYS, CHM) that are affected in patients might help understand the importance of that gene better.
Response
Thank you for the comment. The description for the normal function of each gene has been added in the results section. Please see the updated results part of each gene.

Reviewer 2 Report
The following is my view on the paper "Novel pathogenic mutations identified from whole-genome se- quencing of unsolved patients affected with inherited retinal diseases"
This study aimed to use whole-genome sequencing (WGS) to identify the genetic causes of inherited retinal diseases (IRDs) in a cohort of 311 patients whose mutations had previously remained undetermined. The results of the study found nine putative pathogenic mutations in six IRD patients, including six novel mutations. Four of these mutations were deep intronic, affecting mRNA splicing, while the other five affected protein-coding sequences.
Overall, the study suggests that the use of WGS can enhance the resolution of unsolved cases of IRDs that are not detected through targeted gene panels and whole-exome sequencing (WES). However, the authors note that the overall improvement may be limited. This research highlights the potential of WGS as a valuable tool in identifying the genetic causes of IRDs and providing insight into the underlying mechanisms of these diseases.
The author can compose two or more paragraphs on upcoming suggestions/plans of further experiment verifications.
Author Response
Dear Miss Mara Pop,
Thank you for the review and the opportunity to revise the manuscript submission ID: 2193605 entitled “Novel pathogenic mutations identified from whole-genome sequencing of unsolved patients affected with inherited retinal diseases
”. We greatly appreciate the effort and time expended by the reviewers to provide us with valuable input to improve our manuscript. We have addressed every comment provided by the reviewers. Below, please find the comments and our responses to each comment.
All changes made to the main text were marked by tracked changes in the “Revision_1-30-2023.docx” files.
Reviewer 2’s comments
The following is my view on the paper "Novel pathogenic mutations identified from whole-genome se- quencing of unsolved patients affected with inherited retinal diseases"
This study aimed to use whole-genome sequencing (WGS) to identify the genetic causes of inherited retinal diseases (IRDs) in a cohort of 311 patients whose mutations had previously remained undetermined. The results of the study found nine putative pathogenic mutations in six IRD patients, including six novel mutations. Four of these mutations were deep intronic, affecting mRNA splicing, while the other five affected protein-coding sequences.
Overall, the study suggests that the use of WGS can enhance the resolution of unsolved cases of IRDs that are not detected through targeted gene panels and whole-exome sequencing (WES). However, the authors note that the overall improvement may be limited. This research highlights the potential of WGS as a valuable tool in identifying the genetic causes of IRDs and providing insight into the underlying mechanisms of these diseases.
Comment 1
The author can compose two or more paragraphs on upcoming suggestions/plans of further experiment verifications.
Response
Thank you for your suggestion. We have added an excerpt in the discussion for upcoming suggestions, as well as experimental verification and its implementation for accurate genetic diagnosis and gene based therapeutics in IRD unsolved patients.
